# Chitosan-TiO_2_: A Versatile Hybrid Composite

**DOI:** 10.3390/ma13040811

**Published:** 2020-02-11

**Authors:** Luis Miguel Anaya-Esparza, José Martín Ruvalcaba-Gómez, Claudia Ivette Maytorena-Verdugo, Napoleón González-Silva, Rafael Romero-Toledo, Selene Aguilera-Aguirre, Alejandro Pérez-Larios, Efigenia Montalvo-González

**Affiliations:** 1Laboratorio Integral de Investigación en Alimentos, Tecnológico Nacional de México-Instituto Tecnológico de Tepic. Av. Tecnológico 2595 Fracc. Lagos del Country, Tepic 63175, Mexico; l_m_ae@hotmail.com (L.M.A.-E.); seleagui@gmail.com (S.A.-A.); 2Universidad de Guadalajara, Centro Universitario de los Altos, División de Ciencias Agropecuarias e Ingenierías, Laboratorio de Materiales, Agua y Energía, Av. Rafael Casillas Aceves 1200, Tepatitlán de Morelos 47600, Mexico; napoleon.gonzalez@cualtos.udg.mx; 3Campo Experimental Centro Altos de Jalisco, Instituto Nacional de Investigaciones Forestales, Agrícolas y Pecuarias, Boulevard de la Biodiversidad 2470, Tepatitlán de Morelos 47600, Mexico; ruvalcaba.josemartin@gmail.com; 4Laboratorio de Acuacultura Tropical, División Académica de Ciencia Biológicas (DACBiol)-Universidad Juárez Autónoma de Tabasco (UJAT), Carretera Villahermosa-Cárdenas Km. 0.5 S/N, Rancheria Emiliano Zapata, Villahermosa 86150, Mexico; clau.maytorena@gmail.com; 5Universidad de Guanajuato, División de Ciencias Naturales y Exactas, Campus Guanajuato, Noria Alta S/N, Noria Alta, Guanajuato CP 36050, Mexico; r.romerotoledo@ugto.mx

**Keywords:** chitosan, titanium dioxide, functionalization, hybrid composite, biological activities

## Abstract

In recent years, a strong interest has emerged in hybrid composites and their potential uses, especially in chitosan–titanium dioxide (CS–TiO_2_) composites, which have interesting technological properties and applications. This review describes the reported advantages and limitations of the functionalization of chitosan by adding TiO_2_ nanoparticles. Their effects on structural, textural, thermal, optical, mechanical, and vapor barrier properties and their biodegradability are also discussed. Evidence shows that the incorporation of TiO_2_ onto the CS matrix improves all the above properties in a dose-dependent manner. Nonetheless, the CS–TiO_2_ composite exhibits great potential applications including antimicrobial activity against bacteria and fungi; UV-barrier properties when it is used for packaging and textile purposes; environmental applications for removal of heavy metal ions and degradation of diverse water pollutants; biomedical applications as a wound-healing material, drug delivery system, or by the development of biosensors. Furthermore, no cytotoxic effects of CS–TiO_2_ have been reported on different cell lines, which supports their use for food and biomedical applications. Moreover, CS–TiO_2_ has also been used as an anti-corrosive material. However, the development of suitable protocols for CS–TiO_2_ composite preparation is mandatory for industrial-scale implementation.

## 1. Introduction

In recent years, the development of hybrid composites has increased considerably because they exhibit a wide range of applications, including photo-catalysis, water treatment, and antimicrobial properties [1,2]. Hybrid composites can be made by combining organic–organic (e.g., chitosan–alginate, chitosan–protein, and chitosan–starch), organic–inorganic (e.g., chitosan–TiO_2_, alginate–TiO_2_, and starch–TiO_2_), and inorganic–inorganic (e.g., TiO_2_–ZnO, TiO_2_–MgO, and TiO_2_–Ag) compounds, and they are synthesized by different methods (intercalation of the polymer, sol-gel, hydrothermal, electro–deposition, chemical and physical vapor deposition, suspension and liquid phase deposition). These methods are effective to enhance the technological and mechanical properties of each individual component and also reveal new functionalities [1,3]. Currently, there is a special interest in combining natural polymers such as chitosan (CS) with inorganic materials like titanium dioxide (TiO_2_) to obtain hybrid composites (CS–TiO_2_) with beneficial properties [3,4,5,6].

Chitosan (CS) is a natural biopolymer (linear polysaccharide comprising 1–4 linked 2–amino–deoxy–β–D–glucan) generally obtained by deacetylation of chitin, the main structural component of crustacean exoskeletons. CS exhibits a poly–cationic character and is non–toxic and biodegradable [7]. CS is considered a biological functional compound with multiple interesting properties. It can form films for food and pharmaceutical applications, including edible coatings, packaging material, or as drug–eluting carrier [5,7]. Its adsorbent capacity can have environmental applications during photocatalytic processes of waste–water treatment [8], and it also has inherent antibacterial and antifungal properties [9]. It is biocompatible with several organic and inorganic compounds by the presence of free amino and hydroxyl functional groups in its structure, which can react with other functional groups by electrostatic forces, hydrogen bonds, or by compound–soak up into the polymeric matrix, thus improving its mechanical and biological properties [10]. Moreover, it has been reported that inorganic compounds like TiO_2_, can enhance the mechanical, physical, and biological properties of this biopolymer [5,11].

Titanium dioxide (TiO_2_) is a versatile and chemically inert material with many relevant applications (food, pharmaceutical, biomedical, antimicrobial agent, environmental and clean energy), and the most common forms are rutile and anatase [12]. The wide use of TiO_2_ is supported by its physicochemical, mechanical, and photocatalytic properties; reactivity and thermal stability; and low cost, safe production, and biocompatibility [13]. TiO_2_ can be doped with other inorganic compounds (mainly metallic compounds) to enhance the interfacial charge transfer and restricts the electron–hole recombination, which improve the photocatalytic properties of TiO_2_ under visible light [14]; however, one of the major disadvantages of nano–TiO_2_ is its ability to agglomerate. It has been reported that the interaction between nano–TiO_2_ and biopolymers (starch, gums, and chitosan) can help to reduce the spontaneous agglomeration of the TiO_2_, thus enhancing the functional properties of composite [15].

In the last decade, a strong interest has emerged in hybrid composites and their potential uses, especially in the CS–TiO_2_ composites, which exhibit interesting properties mainly by their combined and additive characters of organic and inorganic materials [1,6]. Behera et al. [4] developed a CS–TiO_2_ membrane with anti–proliferative and antimicrobial properties. Kaewklin et al. [5] fabricated a CS–TiO_2_ film for tomatoes preservation. Saravanan et al. [16] reported that CS–TiO_2_ composites showed great degradation efficiency against methyl orange dye, while Razzaz et al. [17] found that CS functionalized with TiO_2_ nanoparticles exhibited adsorbent properties for wastewater treatment. Nonetheless, Díaz–Visurraga et al. [15] highlighted that size, surface area and mesoporous properties of nano–TiO_2_ enhanced the CS–TiO_2_ interactions, which exhibited great potential industrial applications.

This review summarizes advantages and limitations that functionalization of CS with TiO_2_ nanoparticles offer and provides an overview of the effect of this hybrid composite on the antimicrobial activity and waste–water treatment, potential food–wise, pharmaceutical, and biological applications.

## 2. Hybrid Composites

Immobilization of inorganic compounds onto supporting materials (organic compounds) has increased considerably in the last years, mainly by the development of hybrid composites, which are commonly prepared using protein, starch, gums and chitosan in combination with ZnO, Ag, CuO, and TiO_2_ nanoparticles [1,18,19,20,21]. According to Miyazaki et al. [22], the most common methods for organic–inorganic hybrid–based composite preparation are mechanical stirring (Figure 1a), which is made by simple metal trapped on polymeric matrix (may include chemical interactions between components) where the technological properties are related with the mixing ratio of the polymer and metal oxide; the coating method (Figure 1b), where the metal oxide is deposited in the polymeric surface matrix by electrochemical and/or physical deposition; and the nanocomposite method (Figure 1c), where the polymer and metal oxide are chemically bonded. Nonetheless, the metal oxide can act as crosslinking agent, reinforcing the mechanical properties of polymeric and/or film-forming solution (Figure 1).

TiO_2_ can be incorporated to different organic matrixes as corn and potato starches, carboxymethyl cellulose, sesame, whey proteins, wheat gluten, cellulose, poly-lactic acid, xanthan and gellan gums, and chitosan (alone or in combination) aimed to form hybrid composites with potential industrial applications, as shown in Table 1.

Amin et al. [23] reported that incorporating TiO_2_ into corn starch films improved their mechanical (tensile strength) and thermal (glass transition temperature) properties. According to Sreekumar et al. [35], the presence of TiO_2_ in corn starch films can act as a reinforcement agent, improving its mechanical (tensile strength) and physicochemical properties (degree of swelling in water). Similarly, a maize starch/PVA composite film with TiO_2_ incorporated had an increase of 100% in the tensile strength and water vapor permeability (50% decrease), but these effects were dose-dependent with an optimum TiO_2_ concentration of 0.05% wt.—the film also exhibited UV protection [26]. In potato starch films, the functional properties (tensile strength and elongation at break and moisture content) were improved by the incorporation of TiO_2_ nanoparticles [33]. Besides, it has been reported that Bi_2_WO_6_-TiO_2_/starch nanocomposite films exhibited good ethylene degradation (in vitro) in the presence of UV irradiation [30]. Similar trends were reported when TiO_2_ was incorporated into a combined starch–CS composite by González–Calderon et al. [27] and Li et al. [28], who found that incorporation of nano-TiO_2_ on a starch–CS films improves the mechanical, physicochemical, thermal, and UV barrier properties of the composite. Most of the positive effects promoted by the incorporation of TiO_2_ on the starch coatings/films have been attributed to the strong interfacial interaction between TiO_2_ and the amorphous region of the starch chain [23]. However, it must be considered that an excessive concentration of nano–TiO_2_ on the biopolymer matrix could cause the aggregation of inorganic particles onto the surface of the composite, thus affecting the mechanical and vapor-related properties [26].

Ahmadi et al. [24] fabricated a carboxymethyl cellulose film containing miswak (*Salvadora persica* L.) extract and TiO_2_ nanoparticles. The authors stated that the presence of TiO_2_ (2% wt.) exhibited a marked positive effect on the mechanical and physical properties of the film, it showed enhanced UV barrier properties and antimicrobial activity against *S. aureus* and *E. coli*. Similarly, Balasubramanian et al. [31] found that incorporating TiO_2_ enhanced the elastic, stretchable, and UV protective properties of films formed by a mixture of K–carrageenan, xanthan gum, and gellan gum; it exhibited partial microbial activity against *S. aureus*, but its good thermal stability was mainly attributed to the incorporation of TiO_2_, which can act as a physical cross–linker agent. Furthermore, El–Wakil et al. [11] developed a composite by a casting/evaporation method using a mixture of wheat gluten, cellulose nanocrystals, and TiO_2_ nanoparticles for a potential food packaging material. They reported that mechanical properties (tensile strength and young modulus) were enhanced by the presence of TiO_2_ (0.6% w/w), and the final composite exhibited antimicrobial activity against *E. coli* and *S. aureus*. Nonetheless, the incorporation of TiO_2_ on a hydrophilic polyurethane film provided antimicrobial activity against *E. coli* and *C. albicans* [39].

Fathi et al. [25] develop a nanocomposite film based on sesame proteins incorporated with TiO_2_ nanoparticles and reported that the mechanical properties as tensile strength, water vapor transmission, and water solubility were influenced by theTiO_2_ content as occurring with the starch–based composites with TiO_2_ added, and suggested that the TiO_2_ content should not exceed 3% w/w in the whole composite to obtain good mechanical properties in the protein–TiO_2_ film. This behavior was previously evidenced by Zhou et al. [38], who informed that the addition of small amounts of TiO_2_ (<1% wt.) improved the tensile properties of whey protein films, but decreased the moisture barrier properties, which may be attributed to the interactions between TiO_2_ and whey protein through electrostatic interactions; with high amounts of TiO_2_ (>1% wt.) the moisture barrier properties increased but tensile strength decreased, a possible reason for this behavior is related to the aggregation of TiO_2_ in the whey protein matrix. Furthermore, it has been reported that sesame protein isolate–TiO_2_ composite exhibited photo–degradation (120 min of UV treatment) of methylene blue dye (72%), which is a persistent organic water pollutant [25]. Similar effects were observed by Zheng et al. [34] using an amylose–hallosyte–TiO_2_ composite for methylene blue dye (90%) degradation under UV–irradiation, also adsorptive properties of the composite were enhanced by including TiO_2_. Hammad et al. [29] and Gjipalaj and Alessandri [32] mentioned that cellulose–TiO_2_ and alginate–TiO_2_ are a feasible alternative for the removal of water pollutants like orange–G, methyl orange, and methylene blue dyes due to its adsorptive and photocatalytic properties.

Buzarovska and Grozdanov [37] developed a biodegradable poly(L–lactic acid) (PLA)/TiO_2_ nanocomposite and evaluated their thermal properties and potential use as drug delivery system via in vitro photo, hydrolytic, and enzymatic degradation. As expected, the thermal properties of the composite were enhanced by the presence of TiO_2_ but in a concentration–dependent manner. Similarly, the photo and hydrolytic degradation exhibited an important influence of the TiO_2_ content. Nonetheless, the degradation of the PLA–TiO_2_ composite was faster than pure PLA; however, the opposite effect was observed on enzymatic degradation, suggesting that nanoparticles help with the diffusion of large molecules, such as α–amylase. On the other hand, Rani et al. [36] mentioned that an alginate–TiO_2_ needle composite exhibited a potential use for a tissue engineering application. The fabricated composite exhibited no sign of toxicity on osteosarcoma, fibroblast and human mesenchymal stem cell lines, with well controlled swelling and degradation in comparison with control scaffold.

In general, the incorporation of TiO_2_ improves the physicochemical, mechanical and thermal characteristics of films-based protein and/or polysaccharides, and promote antimicrobial and UV barrier properties, giving them a great potential for food, pharmaceutical or biomedical, and environmental applications.

## 3. Chitosan–TiO_2_ (CS–TiO_2_) Composite

The CS–TiO_2_ hybrid composite is recognized as a promising material with a wide range of applications, mainly by its structural, textural, mechanical, and optical properties; its capacity to function as a water and oxygen barrier; and by its thermal stability and biodegradability. In this context, the effect of TiO_2_ incorporation on a CS matrix on the above properties has been extensively studied (Table 2).

In general, the structural properties of CS are improved by the TiO_2_ incorporation, exhibiting good dispersion and uniformity on the surface of the obtained CS films by becoming rougher and more porous [40,41], but it promotes a slight increase in thickness (70 µm) compared with CS (66 µm). On the other hand, the crystalline peak of CS (2θ = 20.40°) was broader by the effect of TiO_2_, which resulted in a decrease in the crystallinity of the CS–TiO_2_ composite, indicating a major interaction between TiO_2_ and CS [40]. Similar results were observed during the ultraviolet-visible (UV-Vis) spectroscopy studies, where the CS–TiO_2_ exhibited a strong absorption band at 300–500 nm (with CS, no absorbance occurred at 350–800 nm) promoted by the incorporation of TiO_2_ in the composite through hydrogen bonding [42]. Furthermore, the CS–Ag–TiO_2_ and CS–halloysite–TiO_2_ composites showed good stability (zeta potential 31–39 mV) during storage [43,44]. The CS–TiO_2_ composite showed a type II isotherm indicating that the material is macroporous in nature [45]. Nonetheless, the textural (specific surface area (SSA), pore volume, and pore size) properties of the CS–TiO_2_ composite were dose–dependent influenced by TiO_2_ (e.g., major concentration of TiO_2_ promotes an increase in SSA in the hybrid composite) [44,45,46,47]. These changes in structural and textural properties enable CS–TiO_2_ composites to be a good candidate for photocatalysis studies.

The thermal stability of CS–TiO_2_ composites has been assessed through DSC/TGA studies. Li et al. [9] reported major thermal stability of CS–TiO_2_ composite film because the presence of TiO_2_ possibly prevents the thermal decomposition of the composite, mainly by intramolecular interactions [47]. Similar findings were reported by Habiba et al. [48] and Archana et al. [49], who stated that CS started to decompose at 200 °C, while CS–PVA–TiO_2_ started to decompose at 250 °C. This behavior was attributed to the presence of TiO_2_. Moreover, Zhang et al. [50] reported an increase in the glass transition temperature (Tg = 111.13 °C) and enthalpy (318.8 J/g) values in CS–whey protein film (104.46 °C and 258 J/g, respectively) by the incorporation of TiO_2_ [48,49,50,51,52,53]. According to Vallejo–Montesinos et al. [52] smaller amounts of TiO_2_ may promote endothermic transitions to become a narrower composite, while higher amounts of TiO_2_ may produce aggregates in the composite surface, which can affect their thermal stability [53].

Optical properties (opacity and light transmittance) indicate how much light that passes through a material [30]. It has been reported that TiO_2_ promotes a marked change in color, opacity and light transmittance of CS–based composites. [52]. Díaz–Visurraga et al. [15] reported that the film–forming solution of chitosan–TiO_2_ turned whiter when TiO_2_ was added, while the films presented a reduced optical transmittance (brightness and transparency decreased, while opacity increased) in a dose-dependent response [28,30,40,53,54], these changes may be linked to their inherent characteristics and optical properties of TiO_2_ [54]. These optical transmittance modifications have been exploited for the development of packaging and textile industrial applications for their antimicrobial and UV–barrier effect [5,55,56].

The functionalization of the CS matrix through the incorporation of TiO_2_ has gained attention since it can impart the CS–TiO_2_ composite with an enhancement of its mechanical properties [27] because it can act as a reinforcing agent [26], with a slightly increased film thickness (70 µm) compared with CS (66 µm) [57]. Amin and Panhius [23] reported that TiO_2_ (30% vs. 70% CS) significantly improves the mechanical properties (an increase of Young’s modulus in 11.8–fold, a six–fold increase for both tensile strength and toughness) of the CS film. Similar trends were reported during the evaluation of CS–TiO_2_ mechanical properties [27,58,59,60,61]. Al-Sagheer and Merchant [62] mentioned that the viscoelastic properties for hybrid composites, particularly CS/TiO_2_-based composites, depend on the bonding between the polymer–particle interfaces. However, an excessive TiO_2_ content may promote negative effects on mechanical properties of the CS–TiO_2_ composite [27,40], mainly attributed to the inhomogeneous dispersion and agglomeration of TiO_2_ in the film, which cause irregularities in the surface structure of the composite, leading to a reduction of the mechanical properties [60,61,62,63]. The use of TiO_2_ as a CS reinforcing agent has been extensively applied for the development of food packaging [5,40] and wound healing [4,59] materials, as well as for dye removal and photocatalytic applications [16,64].

Water vapor permeability (WVP) and oxygen vapor permeability (OVP), as well as water solubility, swelling capacity, and biodegradability rates, are the most important properties that should be considered during the development of packaging and/or wound–healing materials for food and biomedical applications [40,57,59,60]. In general, water and oxygen permeability values of CS films depend on various interrelated factors such as the structural characteristics of the polymeric chains, hydrogen bonding features and other intermolecular interactions, degree of cross–linking, and dispersion of nanoparticles into the polymeric matrix [1]. Delgado–Alvarado et al. [65] reported that the functionalization of a CS matrix with TiO_2_ nanoparticles reduces the WVP of CS–TiO_2_ films. These results are in line with those of Siripatrawan and Kaewklin [40] and Qu et al. [55] in CS–TiO_2_ and zein–CS–TiO_2_ films. The authors suggested that TiO_2_ may promote a modification of the CS–TiO_2_ film matrix, forming a dense structure, which act as obstacles to water vapor diffusion through the film [28,50]. Similar trends were observed in the OVP properties through the incorporation of nano–TiO_2_ to CS films according to Peng et al. [59] and Lian et al. [60], who stated that the presence of TiO_2_ significantly improves the film functionality by reducing the exchange of oxygen through the CS–TiO_2_ composite film [66]. Besides, no significant differences in water solubility were observed between CS and CS–TiO_2_ films [57]. The presence of TiO_2_ reduces the swelling degree of CS–TiO_2_ in a dose–dependent response [49,63]. Alex et al. [67] reported that TiO_2_–CS–chondroitin 4–sulphate composite showed a maximum swelling ratio of 28%. These results are similar to those reported by Kavitha et al. [68], who observed no significant changes in the swelling capacity above a 2:1 (CS:TiO_2_) ratio because TiO_2_ can control the swelling capacity of CS–TiO_2_ composite [47] and moderately prevents the mass loss of the composite [51]. According to Amin and Panhuis [23], the swelling decrease in CS–TiO_2_ films could be attributed to the presence of TiO_2_ into the composite and by its ability to participate in hydrogen bonding with CS. Furthermore, CS–TiO_2_ exhibited moderate biodegradability (<3%), and favorable time-dependent biodegradability (0.0521 mg mL^−^^1^ h^−^^1^ on the first day; 0.0006 mg mL^−^^1^ h^−^^1^ post-covering) [59]. In this context, the CS–TiO_2_ composite exhibits great potential for food and biomedical applications [40,59].

Additionally, the structural changes of CS by the presence of nano-TiO_2_ has been extensively evaluated by Fourier transform infrared (FTIR) studies. Figure 2a shows a typical FTIR spectra of a CS–TiO_2_ hybrid composite (consistent with reported FTIR spectra in literature for CS–TiO_2_ composites [16,40,48]), where the most representative FTIR band positions of the chitosan–TiO_2_ hybrid composite are listed in Table 3.

According to Siripatrawan and Kaewklin [40], the incorporation of TiO_2_ into the chitosan matrix mainly occurs in the amorphous region of chitosan, strengthening the hydrogen bond (3300 cm^−1^) in the CS–TiO_2_ complex [69,71] (Figure 2a). It has been reported that the electron doublet that is free on the nitrogen atom (–NH_2_) and –OH functional groups on chitosan structure is the responsible for interaction and may serve as a coordination bond (3450 cm^−1^) and reaction site for the adsorption of transition metals and/or other compounds [8,76] (Figure 2b). Saravanan et al. [16] suggested that the stretching vibrations of C–O, amino, and hydroxyl groups are strongly attached to TiO_2_ nanoparticles which promote the formation of CS–TiO_2_ composite via electrostatic interaction of N–H–O–Ti bonds (3350 cm^−1^). Similarly, Kavitha et al. [70] detected O–Ti–O bonds formed between TiO_2_ and CS (385–900 cm^−1^). In addition, a new band appeared at 1000 cm^−1^ which corresponds to the Ti–O–C bond, demonstrating that CS and TiO_2_ have been chemically bonded and not only adsorbed into the CS matrix [75] as a result of a crosslinking process and the complex composite structure [60,77]. Nonetheless, during the CS–TiO_2_ preparation, the tendency of TiO_2_ to agglomerate is prevented because the chitosan masked the Van der Waals influence on TiO_2_ nanoparticles [15,70], due to the immobilization of nano-TiO_2_ in the chitosan matrix [41].

Raut et al. [10] mentioned that the incorporation of TiO_2_:Cu-doped did no altered the characteristic structure of CS and reported that the metal–oxygen–metal inorganic union was bonded with CS macromolecules by hydrogen bonding as well as covalent bonding in the CS–TiO_2_:Cu composite; however, in the same study, Raut et al. [10] informed about some minimal changes in the other peaks using X-ray photoelectron spectroscopy studies, which manifest new covalent bonds formation. Zhang et al. [50] reported that CS–TiO_2_ exhibited a shift of 21 cm^−^^1^ compared to CS (at 1631 cm^−^^1^), confirming the interaction between CS and TiO_2_ nanoparticles [74]. On the other hand, it has been reported the presence of titanates (by the effect of acidic conditions during preparation) in CS–TiO_2_ composite (1735 cm^−^^1^) [71]. This fact may promote a decrease in the crystallinity of the CS matrix (decrease in intensity of –OH and –NH bands), probably to the effect of matrix alteration and interfacial interaction between organic and inorganic phases [5]. However, all effects of TiO_2_ on the CS structure are dependent of factors as size and concentration of TiO_2_ nanoparticles [16,42,55], but also by the available free amino group resulting for the interaction of TiO_2_ with CS [76]. However, the creation of a stable CS–TiO_2_ composite is CS and TiO_2_ concentration and pH dependent (favorable pH 4.8) [51].

## 4. Applications of CS–TiO_2_ Composite

The CS–TiO_2_ hybrid composite has been widely used for various technological applications such as antimicrobial agents, packaging, films and/or coating materials for food preservation, wound healing and skin regeneration, glucose, α–fetoprotein detection, and water pollutant degradation due to their prospective benefits, as discussed below. Figure 3 illustrated the main applications of CS–TiO_2_ hybrid composite, which include photocatalytic properties and barrier, antimicrobial, and wastewater treatment applications.

### 4.1. Antimicrobial Activity

CS and TiO_2_ NPs exhibit antimicrobial activity against bacteria, yeast, and molds [4,40,78,79], which could be enhanced by their combination, via CS–TiO_2_ composite (Table 4).

It has been reported that the antimicrobial activity of CS was improved by the incorporation of TiO_2_ into a polymeric matrix [4,54,74], and a similar behavior was reported using a combination of CS with zein (Zein/CS) [55], and carboxymethyl cellulose (CS/CMM) [64] adding nano-TiO_2_. Li et al. [9] reported that CS–TiO_2_ composites exhibited a higher antimicrobial activity against *Xanthomonas oryzae pv. Oryzae* than the two individual components under light and dark conditions. Longo et al. [82] explained that TiO_2_ could be present or trapped on the composite surface, thus enhancing their antimicrobial activity due to the generation of electron–hole pairs without the need of light–induced radiation (Figure 3b). Similar results were reported by Li et al. [81] and Xiao et al. [6] using a CS–Ag–TiO_2_ composite against *E. coli*, *S. aureus*, *P. aureginosa*, and *C. albicans,* probably for the synergistic effect of the three components, which suggests that microbial inactivation is linked with electrostatic interactions between the cell membranes and the composite, followed by oxidative stress caused by photo–generated reactive radicals, causing cell death (Figure 3d).

Siripatrawan and Kaewklin [40] observed that CS–TiO_2_ nanocomposite films exhibited antibacterial activity against Gram–negative (*E. coli*) and Gram–positive (*S. aureus*) bacteria and fungi (*Aspergillus* and *Penicillium*). Similarly, Díaz–Visurraga et al. [15] prepared a semitransparent composite film using CS and TiO_2_ nanotubes by casting plate. They highlighted that the hybrid composite exhibited antimicrobial activity against *S. enterica*, *E. coli*, and *S. aureus*; where the antimicrobial effect was strain– and TiO_2_ concentration–dependent. Additionally, studies using transmission electron microscopy (TEM), have revealed that CS–TiO_2_ films promote bacterial cell disruption, membrane deformation and structural changes of the cell wall. Kavitha et al. [68] mentioned that the CS–TiO_2_ film exhibited moderate antibacterial activity against *S. aureus*, but no against *E. coli* and *K. pneumoniae*. On the other hand, Kim et al. [78] found that when CS–TiO_2_ composites are used, the death cell ratio depends on several factors (pH, temperature, UV light, and TiO_2_ concentration) and their potential interactions. They reported that, at constant TiO_2_ concentration, the death ratio of *S. choleraesuis* increased when pH and temperature decrease, and, at low pH values, the death ratio increased when TiO_2_ concentration increased. The antimicrobial activity is affected by the TiO_2_ concentration even in the presence of UV–radiation (30 min). At concentrations of 1 mg TiO_2_ mL^−^^1^, 48% of *S. choleraesuis* lost cell viability, while at 0.25 mg mL^−^^1^, the killing efficiency decreased by 22%.

Furthermore, the antimicrobial activity of CS–TiO_2_ composites could be enhanced by the modification of the TiO_2_ band gap associated with the presence of other compounds in the TiO_2_ matrix [14]. Raut et al. [10] informed that the incorporation of TiO_2_:Cu–doped nanoparticles (1 mg mL^−^^1^) into CS matrix (CS–TiO_2_:Cu) increased up to 200% the photocatalytic antimicrobial activity of CS–TiO_2_ composite against *E. coli* and *S. aureus*. After 3 h of treatment, more than 85% of the bacteria cells lose their viability, and the presence of Cu in TiO_2_ nanoparticles decrease the band gap of TiO_2_^,^ and reduced the recombination of electron–hole pair rate, resulting in a better photocatalytic performance through Fenton chemistry. Similar trends were observed by Li et al. [72] against *E. coli*, *S. aureus* and *P. aureginosa* using a CS–TiO_2_:Ag composite. They concluded that the increase of antibacterial activity is attributable to the increase of the surface area of TiO_2_ by silver doping. Kamal et al. [64] suggested that the antibacterial mechanism of the hybrid composites could be associated with the interaction of electrostatic charges between CS–TiO_2_ (positive) and bactericidal membranes (negative), which may promote a membrane cell alteration, blocking nutrient intake and affecting the viability and normal cell growth.

Shi et al. [79] reported that gauzes impregnated with a CS–TiO_2_ emulsion were effective to inhibit the growth of *E. coli* (99.9%), *A. niger* (100%), and *C. albicans* (78.3%). The gauzes could also be reused up to eight times without loss of this antibacterial ability. Similar trends were reported by Xiao et al. [41], who observed that a CS–Fe–TiO_2_ composite coating showed favorable shelf life (up to one year) without losing its antibacterial (*E. coli*) and antifungal (*C. albicans* and *A. niger*) activities. Qian et al. [42] found that the incorporation of CS–TiO_2_ composite on cotton fibers exhibited antimicrobial activity against *E. coli* (99.9%), *S. aureus* (99.9%), and *A. niger* (97.4%). Moreover, Xu et al. [80] observed that *A. niger* is less susceptible to a graphene oxide (GO)–CS–TiO_2_ composite than *B. subtilis* at the same experimental conditions. They also stated that a GO–CS–TiO_2_ composite can selectively increase the permeability of microbial cell membranes, promoting a leakage of cell contents, mainly by the direct physical interaction between the GO–CS–TiO_2_ composite and the microorganisms. Additionally, it has been reported that nano–TiO_2_/CS/PVA ternary composite exhibited enhanced antimicrobial activity against *E. coli* and *S. aureus* than the individual components [81].

The CS–TiO_2_ composite exhibit enhanced antimicrobial activity against Gram–positive and Gram–negative bacteria, yeast, and molds, which are favorable in many industrial domains such as the food industry for developing food packaging material and for biomedical applications such as wound–healing material.

### 4.2. Environmental Applications

Recently, a growing number of organic pollutants have been discharged into all kinds of open waters [16]. Therefore, great emphasis has been put on water treatment, mainly regarding pollutant degradation [64]. Heterogeneous photocatalysis based on semiconductor photocatalysts is a viable alternative because it is an environmentally friendly and cost–effective method for the water pollutant degradation [74]. Nonetheless, in recent works, researches have demonstrated that the incorporation of TiO_2_ into a polymeric matrix, such as CS and thus forming a hybrid composite (CS–TiO_2_), provided a major stabilization of the semiconductor material (TiO_2_), which resulted in higher pollutant degradation than the individual components. This may be attributable to the adsorbent and chelating properties of CS, which exhibited great potential for water treatment, and to the photocatalytic properties of nano–TiO_2_ [83] as illustrated in Figure 3e. Table 5 shows the photocatalytic activity of a CS–TiO_2_ composite for different water pollutant degradation.

Zhang et al. [50] synthetized a CS–TiO_2_ composite through a metal–binding reaction using glutaraldehyde as a crosslinking agent and used it for the detoxification of hexavalent chromium from water. The CS–TiO_2_ composite exhibited good adsorption capacity of chromium (171 mg g^−^^1^), and reduced Cr(VI) to Cr(III), a less toxic form of Cr, suggesting that the removal mechanism could involve the Cr(VI) adsorption by electrostatic interactions (Ti^4+^ and HCrO^4−^) and ligand exchanges (Cl^−^ and HCrO^4−^), promoting a Cr(VI) reduction to Cr(III) and re-adsorption of Cr(III) onto CS–TiO_2_ composite. It has also has been reported that oxygen presented in the solution can acts as a “hole scavenger” and generates H_2_O, besides reduced Cr(VI), as reported by Farzana and Meenakshi [75], who found that the removal efficiency of Cr(VI) was initial–concentration dependent—at higher concentrations of Cr(VI) (125 mg L^−^^1^) a reduction of 74% was achieved, while at low initial concentrations, the efficiency increased considerably (96.7%).

Razzaz et al. [17] functionalized CS nanofibers with nano-TiO_2_ by a coating and entrapped method for the removal of heavy metal ions—Cu(II) and Pb(II)—from water. Both CS–TiO_2_ composites exhibited good adsorption capacity for Cu(II) (710 and 579 mg g^−^^1^, respectively) and Pb(II) (526 and 475 mg g^−^^1^, respectively) after 30 min at 45 °C. Nonetheless, the composites prepared by the entrapped method exhibited better results after five adsorption/desorption cycles than the composite prepared by the coating method. The CS–TiO_2_ composite showed selective sorption in order of Cu(II) > Pb(II), and the authors discuss that the adsorption effectiveness is related to the metal ions concentration and the adsorption ion capacity of the hybrid composite. Furthermore, it has been reported that the ternary CS–hemicellulose–TiO_2_ composite exhibited enhanced properties for the adsorption of heavy metals ions as Ni(II), Cd(II), Cu(II), Hg(II), Mn(II), and Cr(VI) from an aqueous solution due to chelating groups present in the structure, without significant loss of its adsorptive capacity up to five cycles [90]. Similarly, Alizadeh et al. [93], using a cross–linked magnetic EDTA/CS/TiO_2_ composite, reported a maximum adsorption capacity of Cd(II) of 209 mg g^−^^1^ and phenol degradation efficiency of up to 90% after up to five cycles of reuse. The process of adsorbing Cd(II) on EDTA/CS/TiO_2_ has been associated to the presence of functional groups of CS and a synergistic effect with cross–linked magnetic EDTA/TiO_2_. Moreover, Xie et al. [91] developed an electrode by modifying nano-TiO_2_/CS composite film onto a glassy carbon electrode for the determination of Cd(II) with a detection limit of 2.0 × 10^−^^10^ mol L^−^^1^ Cd for 180 s accumulation, representing a simple, effective, faster and economical alternative method for Cd detection.

CS–TiO_2_ has also been used to oxidize As(III) to As(V) (a less toxic and more easily sequestered arsenic form) in the presence of UV light and oxygen (adsorption capacity of 6.4 mg As(V) g^−^^1^ and 4.92 mg As(III) g^−^^1^) by a hybrid composite [85,96]. In addition, the TiO_2_–enabled CS cross–linked with copper (CS–TiO_2_:Cu-dopped) is an alternative for arsenite and arsenate water removal via photo–oxidation and/or adsorption process, which was mainly associated to the synergistic effect between TiO_2_:Cu loaded into a CS matrix. Furthermore, during the degradation process, CS–TiO_2_:Cu photo–oxidizes As(III) to As(V) through ROS generation, and As(V) is chelated due to the presence of Cu into the composite, which acts as an As(V) chelating agent. Moreover, it has been reported that a CS–TiO_2_:Cu composite exhibited better adsorption properties than traditional adsorbents (Cu, TiO, Al_2_O_3_) for As(III) and As(V) removal in systems buffered at pH 6 [87]. These results are in agreement with those of Yazdani et al. [83] during arsenate (73%) and arsenite (84%) removal from an aqueous solution using a CS–TiO_2_–feldspar composite. They found that UV irradiation increased As(V) removal regardless pH conditions. According to Wu et al. [90], the dispersion stability of the nano–TiO_2_ in the polymeric matrix is a determinant factor to improve the 3D network and surface area of a hybrid composite for metal ions adsorption.

CS–TiO_2_ has an excellent photo–catalytic degradation efficiency against methyl orange (MO) dye [16]. Zhu et al. [84] mentioned that CdS nanocrystals deposited on TiO_2_/crosslinked CS composite (CdS/TiO_2_/CS) is a viable alternative for decolorization treatments of water contaminated with dyes due to its photocatalytic and adsorbent properties. The composite exhibited a good methyl orange adsorption character (MO, 2.66 mg g^−^^1^) and high photocatalytic degradation potential of MO–solution (99.1%) after simulated solar light radiation (210 min). The photocatalytic activity (89%) was maintained after five batch reactions. Moreover, a CdS/TiO_2_/CS composite could be also used for Congo red degradation [77]. Afzal et al. [94] reported that TiO_2_–supported CS exhibited a complete degradation of MO within 90 min of UV exposure. Similarly, Zainal et al. [8] mentioned that during MO-removal (87%) from water using a CS–TiO_2_ composite, the reactive –NH_2_ and –OH functional groups of CS and TiO_2_ are directly responsible for the photodegradation–adsorption process during abatement of various wastewater pollutants. During MO degradation (90%) using CS–TiO_2_ composite, Li et al. [86] found that the MO removal was the result of an additive effect of both CS and TiO_2_ and not only by the adsorptive properties of CS. Furthermore, Xiao et al. [6] reported that a CS–TiO_2_ composite showed enhanced photocatalytic selectivity for MO compared with CS, and the hybrid composite could be reused up to 10 cycles while preserving 60% of its efficiency. According to Zhu et al. [97], a TiO_2_/ZnO/CS composite exhibited high photocatalytic degradation of MO (>97%) solution, which was mainly attributed to the adsorption ability of the composite and its high surface area. In addition, the decolorization process was accelerated under simulated solar irradiation. Zhu et al. [97] mentioned that the addition of ZnO into the TiO_2_ matrix (TiO_2_:ZnO) and then to the CS matrix has a beneficial role in improving the charge separation and extending the response of the composite to visible light. Additionally, it has been reported that a CS–TiO_2_–rGo composite performs well as a MO degrader (97%) [58]. Meanwhile, Dhanya and Aparna [88] reported that CS may be a good and economically support for TiO_2_ immobilization (by hydrogel method), and the resultant composite exhibited a 100% of degradation of MO and alizarin red dyes after 3 h of treatment under UV light (293 nm) irradiation. Furthermore, this photocatalyst can be recycled up to three times without efficiency losses (65%). The efficiency for dye decoloration is pH–dependent (pH 11 < pH 9 < pH 7 < pH 5 < pH3). Similarly, Habiba et al. [71] observed a complete degradation of MO using a Chitosan/PVA/Na–Titanate/TiO_2_ composite under UV irradiation. According to Habiba et al. [71], the deacetylation degree (DD) of CS plays a key role for MO removal, mainly by the available active sites, where CS with high DD exhibited better adsorption capacity for MO removal than medium DD [48].

Haldorai and Shim [74] reported that a CS–TiO_2_ composite was effective for the degradation (90%) of methylene blue (MB) under UV–light illumination without significant losses of catalytic activity after five cycles of reuse. In addition, the obtained composite could be an excellent candidate as an environment-friendly photocatalyst due to its low cost and catalytic efficiency. Similar trends were reported by Nithya and Jothivenkatachalam [45], who observed a complete (100%) degradation of MB (initial concentration 5 mg L^−^^1^) using CS–TiO_2_ composite at 0.3 g L^−^^1^ under UV-light and suggested that MB degradation could be accelerated under alkaline conditions (pH 11).

Chen et al. [73] fabricated a thiourea magnetic ion-imprinted CS–TiO_2_ composite for simultaneous cadmium removal and 2,4–dichlorophenol (2–4–DCP) degradation. The prepared composite exhibited good cadmium adsorption (256 mg g^−^^1^) and 2,4–DCP degradation (98%) capabilities (at initial concentration of 10 mg L^−^^1^). Several intermediates (4-chlorophenol, 1,4–benzoquinone, phenol, and cyclohexanol) were formed during the degradation process in the presence of UV light, and the authors suggested that the major decomposition pathway of 2,4–DCP is linked to the reductive dechlorination and reaction with hydroxyl radicals. Liu et al. [95] evaluated the adsorptive removal of 2,4–DCP from water using a CS/activated carbon fiber (ACF)/TiO_2_ membrane and reported that the membrane possesses high filtration properties (in static and dynamic modes) and high adsorptive removal of 2,4–DCP using both flux modes. Furthermore, the membrane adsorption capacity can be regenerated by oxidizing adsorbed 2,4–DCP via in situ or off–site photocatalysis and/or Fenton oxidation. Xiao et al. [89] incorporated Ag^+^ ions into the surface of CS–TiO_2_ composites and found and enhanced efficient catalytic activity in the reduction of 4–NPh (hazardous material) to 4–APh (commercial compound) by NaBH_4_, with 100% conversion reached in 120 min without catalytic activity losses after five continuous cycles.

Zhou et al. [92] prepared a nano–TiO_2_/CS/poly(N-isopropyl acrylamide) composite hydrogel for the removal of ionic dyes as acid fuchsin (AF). The hydrogel composite exhibited high photocatalytic degradation of AF with a removal rate of 90% under UV radiation after 160 min. However, the effectiveness of the composite was pH– and temperature–dependent, exhibiting a better performance at pH 4 and 20 °C. Authors also reported a decrease in the removal rate of AF when temperature increased (40 °C), which may be associated with the gradual hydrogen bond weakening between amine groups of AF and hydrophilic groups (–NH_2_, –OH, and –CONH–) of the hydrogel composite. According to Longo et al. [82], these reactions can occur under coordinated cationic Ti surface sites, and the reaction efficiency is directly proportional to the number of adsorptive/photoactive sites of the composite [48]. On the other hand, for thymol violet removal from water, a TiO_2_/CS/carboxymethyl cellulose composite seemed to be a promising alternative [64].

In summary, research–based reports support the assertion that CS–TiO_2_ is an attractive material with high performance and a low–cost and is efficient, reusable, and environmentally friendly for heavy metal ion removal and pollutant degradation from water.

### 4.3. Biomedical Applications

Miyazaki et al. [22] highlighted that organic–inorganic composites—and CS–TiO_2_ in particular—exhibited interesting biomedical applications. It has been recognized that the incorporation of TiO_2_ into a CS biopolymer is a strategy to enhance the biological activities of CS [3]. Table 6 shows the potential biomedical applications of a CS–TiO_2_ composite.

Behera et al. [4] evaluated the effect of a CS–TiO_2_ composite membrane on the proliferation and survival of L929 fibroblast cells as potential wound dressing and skin regeneration material. The results showed no cytotoxicity effects on L929 cells growth by the effect of the composite and the CS–TiO_2_ membrane can scavenge the reactive oxygen species (ROS) over–production in the L929 cell line, suggesting that the composite exhibited good antioxidant capability. Furthermore, the cells under CS–TiO_2_ showed a favorable effect on the cell cycle progression (G0/G1 phase), thus promoting the proliferation and survival of fibroblast cells. Peng et al. [59] reported that a CS–TiO_2_ composite showed a reduced effect on the pro–inflammatory TNF–α and anti-inflammatory cytokine IL–6 when comparing the control to commercial products (Duoderm); nonetheless, CS–TiO_2_ exhibited a better wound healing effect than the commercial product during an in vivo trial. Similar results were reported by Archana et al. [49], who observed that CS–pectin–TiO_2_ composite exhibited good in vitro results by inducing blood coagulation and showed good hemostatic properties and no cytotoxic toward L929 or NIH3T3 mouse fibroblast cells. Additionally, the in vivo open excision–type wound healing study in adult male albino rats revealed that CS–pectin–TiO_2_ (99% of closure) had better results than CS (95%) and gauze (91%) after 16 days. Similarly, Woo et al. [101] fabricated a wound dressing material with a bilayer composite of nano–TiO_2_ combined with a CS membrane and a sub–layer of human adipose–derived extracellular matrix (ECM) sheet, and reported that, from an in vivo experiment using rats, the TiO_2_–CS–ECM had faster regeneration of the granulation tissue and epidermis with less scar formation in comparison with control wounds. It has been suggested that the wound–healing effect of the CS–TiO_2_ composite is related to the regulation of re–epithelialization and inflammation process [76].

Zhang et al. [102] reported that the self-assembly CS/gelatin composite coating on icariin(ICA) –modified TiO_2_ nanotubes (by physical adsorption) exhibited osteoblast bioactivity; which can modulate primary osteoblasts due to the ICA–Ti surface facilitating the initial adhesion, promoting osteoblastic proliferation and up–regulating the expression of bone–related genes (osteopontin, type I collagen and osteoprotegerin), while down–regulating RANKL mRNA expression, indicating a potential application on Ti implants to improve biocompatibility and osseointegration. Alex et al. [67] mentioned that TiO_2_–CS–Chondroitin 4–sulphate composite exhibited great potential for bone implant applications because it showed excellent bioactivity, biocompatibility and high cell viability at low concentrations on human MG–63 cell line. Likewise, Safari et al. [103] evaluated the in vitro drug–delivery properties of a CS/dopamine(DOP)/TiO_2_ system (the solution was at pH 7.4) and concluded that the incorporation of TiO_2_ on a CS/DOP system enhanced the drug entrapment, and provided a controlled drug–release effect (16 h) compared with CS/DOP system (10 min). This fact was attributable to the strong hydrogen bonding interactions between CS/DOP systems with –OH groups on the TiO_2_ surface.

Additionally, Huang et al. [98] developed an amperometric immunobiosensor for α–fetoprotein detection (AFP oncofetal glycoprotein, which is a clinical cancer biomarker) using an Au/CS/TiO_2_–graphene composite based platform. The biosensor was a novel strategy for AFP detection with good bioactivity, sensitivity, and selectivity with good storage stability. Nonetheless, an Au/CS/TiO_2_–graphene composite is an interesting alternative to the development of biosensors aimed to the detection of diverse antigens or compounds [104]. Similarly, Al–Mokaram et al. [99] fabricate a non–enzymatic glucose biosensor based on electrochemically prepared polypyrrole(Ppy)–CS–TiO_2_ nanocomposite film and reported that the device showed good sensitivity over a linear range of 1–14 mM and a detection limit of 614 µM of glucose. The biosensor also exhibited high selectivity and extended useful life with no interference effect in comparison with Ppy–CS without TiO_2_, where TiO_2_ increased the electrocatalytic activity of the electrode surface. Furthermore, the Ppy–CS–TiO_2_ nanocomposite can be used as a glucose sensor due to the ability of composite for glucose oxidation [99]. These results are in line with those of Zhang et al. [100], who developed a glucose biosensor on a TiO_2_–multiwall carbon nanotubes–chitosan composite and functionalized with Au nanoparticles, which performed as a good glucose detector with a linear range of 6 µM to 1.2 mM and a detection limit of 0.1 µM. This was attributed to the increase in the surface area of the composite by the presence of TiO_2_ and Au nanoparticles, which facilitated the electron transfer to the electrode surface.

In general, the CS–TiO_2_ composite exhibits great potential for biomedical applications mainly by its low cytotoxic effects on different cell culture lines [4,49,59]. Jayakumar et al. [47] reported that a CS–TiO_2_ composite scaffold did not exhibit cytotoxicity on cell lines such as osteoblast–like cells (MG–63), fibroblast cells (L929), and human mesenchymal stem cells (hMSCs) and highlighted the potential use of a CS–TiO_2_ composite for biomedical applications. Raut et al. [10] observed that CS–TiO_2_:Cu–doped exhibited low toxicity toward mammalian cells with more than 85% of cell viability after 72 h of exposure. Similar results were reported by Xu et al. [80], who observed a low toxicity degree in mammalian somatic and plant cells of GO–CS–TiO_2_ coating. Moreover, Kavitha et al. [46] and Li et al. [9] reported no significant cytotoxic effects of CS–TiO_2_ on gastric carcinoma cell line and L929 cell line using a TiO_2_–CS–PVA composite, respectively. However, the preparation method of hybrid composite has a higher impact on the biocompatibility between CS and TiO_2_, which is linked with its biological activity [46]. Therefore, the development of suitable protocols for CS–TiO_2_ composite preparation is needed in order to optimize the CS–TiO_2_ fabrication.

### 4.4. Food Preservation Applications

Most of the food applications described in the literature for CS–TiO_2_ composites are focused on developing food packaging alternatives and edible coatings for fruit preservation (Table 7). It has been reported that the inclusion of TiO_2_ into food packaging materials could prevent degradation associated with UV light because TiO_2_ acts as a filter against radiation, thus protecting the food inside the package [15], as illustrated in Figure 3c.

Kaewklin et al. [5] developed an active packaging with CS–TiO_2_ nanocomposite aimed to enlarge the shelf life of tomato fruit, founding that tomato packaged in the CS–TiO_2_ film had lower quality changes (firmness, weight loss, color, total soluble solids, lycopene, and ascorbic acid content) than those in the CS control group after 15 days of storage at 20 °C. These results were attributed to the ethylene photo–degradation achieved by the presence of TiO_2_ (UV–irradiated at 320–390 nm during 180 min) into the film and consequently delayed the ripening process and storage–wise changes on quality parameters of tomatoes. Tian et al. [106] mentioned that a CS–TiO_2_ coating film was effective for *Gingko biloba* seeds preservation without a significant reduction of firmness and antioxidant capacity, and also, prevented the mildew apparition and reduced some fruit metabolic processes (respiration rate and ethylene production), which delayed the senescence of gingko seeds, concluding that CS–TiO_2_ film is an attractive method for food preservation. Similar results were reported on ethylene photo–degradation (in vitro and TiO_2_ dose–dependent) by Siripatrawan and Kaewklin [40] and Zhang et al. [54] using a CS–TiO_2_ nanocomposite and CS–TiO_2_–black plum peel extract (BPPE) films, respectively. The authors suggested that the incorporation of BPPE into CS–TiO_2_ composite improves the antioxidant capacity of the resulting packaging material [57].

Zhang et al. [54] reported that a CS–TiO_2_ film was effective to extend the shelf life of red grapes storage at 37 °C up to 15 days in comparison with a CS–based film, proving the improvement of wettability and mechanical properties of CS film through the addition of TiO_2_ nano–powder. Yuan et al. [105] formulated a CS/nano–TiO_2_ composite coating and evaluated their effect in extending the postharvest shelf life of Stauntonvine fruit. The fruit treated with the composite coating exhibited a reduction of the CO_2_ transmission coefficient (27 g d^−^^1^) compared to the CS coating (35 g d^−^^1^); preventing the loss of ascorbic acid and changes in pH, titratable acidity, and total soluble solids after 45 days of storage at room temperature. This behavior was related to the modification on the internal atmosphere (CO_2_ and O_2_ ratio) between the fruit and the composite coating. Likewise, Xu et al. [80] mentioned that a GO–CS–TiO_2_ coating was effective to extend the shelf life of strawberries and mangoes without significant weight loss (<5%), and maintaining good appearance, color, and brightness, which was attributable to the low polyphenol oxidase enzyme activity and the activation of the fruit antioxidant enzymes system, including superoxide dismutase activity. On the other hand, Lin et al. [44] mentioned that a CS–Ag–TiO_2_ coating showed good adherence (an important parameter for edible coatings) when used on cantaloupe rind.

Despite the observation that CS–TiO_2_ exhibited great potential to be used for food packaging purposes (good adherence in food surface, prevented microbial infection, and ethylene scavenging activity), it is necessary to assess the possible TiO_2_ migration from CS–TiO_2_ films into the food matrix [44]. García et al. [107] recently informed that migration studies in other types of hybrid composites (CS–ZnO and CS–Ag composites) have demonstrated that only a negligible amount of the nanomaterial migrates from the package to the food simulants or foods, suggesting that consumer exposure to these nanomaterials and associated health risks are minimal. However, further studies are needed to evaluate the potential migration of TiO_2_ from hybrid composites to food matrices and its toxicity potential.

### 4.5. Other Applications

Table 8 shows other potential applications of CS–TiO_2_, such as anti-metal corrosive material, as well as for the development of textile materials with protective effects against solar radiation.

Chen et al. [56] integrated a CS–TiO_2_ composite into a cotton matrix by a dip–pad–dry cure process for sun blocking applications and reported that the treated cotton fabrics demonstrated lower photocatalytic activity in comparison to cotton fabrics treated with pure TiO_2_ in a typical photocatalytic test. The authors highlighted that the CS–TiO_2_/cotton showed excellent UV protection and suggested that the nanocomposite could be used on materials that are in direct contact with the human body. Similar trends were reported by Hsieh et al. [108] using wool fabrics cured with citric acid and TiO_2_–CS.

Bajali and Sethuraman [109] studied the effect of a CS–doped hybrid/TiO_2_ (1:1 molar relation) nanocomposite by a sol–gel method and a self–assembled method on the corrosion resistance of aluminum in NaCl medium (3.5%) through electrochemical impedance studies and potentiodynamic polarization. The presence of the nanocomposite improved the corrosion resistance of aluminum (efficiency 90%) compared to undoped CS (66%). The aluminum corrosion resistance may be attributable to the formation of a dense network structure formed by the Al¬–O–Ti–O–Si–O–Si bond. Meanwhile, Ledwing et al. [110] achieved the optimization of electrophoretic deposition parameters for manufacturing of TiO_2_/CS composite coating on X2CrNiMo17-12-2 stainless steel. This hybrid coating (4–8 µm thickness film) improved the corrosion resistance of steel (−165 to −110 Ecorr) compared to uncoated steel (−291 Ecorr), and the composite coating exhibited a fully uniform distribution of TiO_2_ (3 g L^−^^1^) nanoparticles on the chitosan (0.5 g L^−^^1^) surface. The increase in the corrosion resistance may be due to a significant reduction in the exposure area between metal and Ringer’s solution. On the other hand, the adhesion of TiO_2_/CS coating on steel was dependent on the CS/TiO_2_ ratio in the coating.

## 5. Conclusions

Evidence shows that the incorporation of TiO_2_ into the chitosan (CS) matrix is a viable way to enhance the technological and biological properties of both compounds. In general, CS–TiO_2_ exhibits better physicochemical, mechanical, and thermal characteristics than pure chitosan, with high biocompatibility and low cytotoxic effects on different cell lines. It is an attractive material with high performance, is low–cost, efficient, reusable, and environmentally friendly for diverse industrial applications based on its antimicrobial and photocatalytic characteristics.

CS–TiO_2_ hybrid composite development is an active research area for biomedical and environment remediation applications and, hence, shows great value for industrial applications. A crescent research area for the application of CS–TiO_2_ is food science and preservation, where their major challenges are the safe implementation of a hybrid composite and their public acceptance. However, additional research is needed to standardize and further develop the CS–TiO_2_ hybrid composites because their technological and biological activities depend on their preparation method.

## Figures and Tables

**Figure 1 materials-13-00811-f001:**
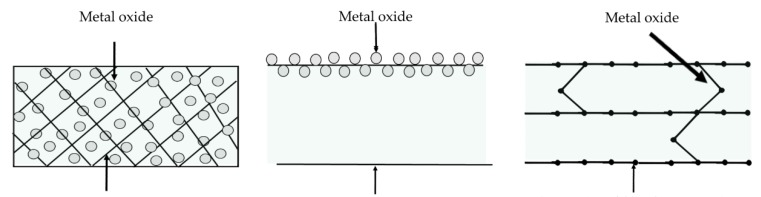
Schematic representation of hybrid composite preparation: (**a**) mechanical stirring, (**b**) coating surface, and (**c**) nanocomposite (adapted from Miyazaki et al. [22]).

**Figure 2 materials-13-00811-f002:**
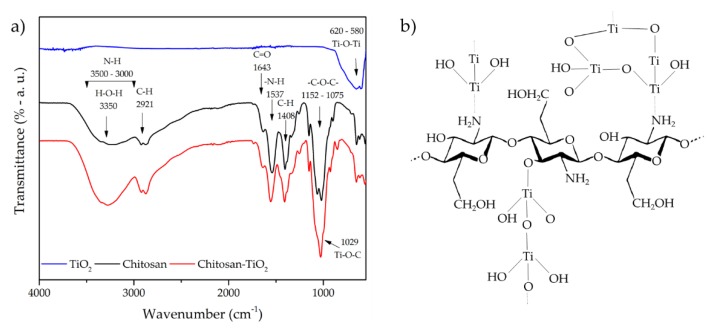
Fourier transform infrared spectra of TiO_2_, chitosan, and chitosan–TiO_2_ hybrid composite (**a**) (the CS–TiO_2_ composite was prepared in our laboratory following the methodology of Siripatrawan and Kaewklin [40] by using 1 g chitosan with medium molecular weight (Sigma-Aldrich, St. Louis, MO, USA) dissolved in 1% acetic acid solution, the nano-TiO_2_ (previously characterized in reference [14]) was 1% of the total chitosan weight, and 15% of glycerol weight of the total solids), and schematic representation of CS–TiO_2_ hybrid composite structure (**b**).

**Figure 3 materials-13-00811-f003:**
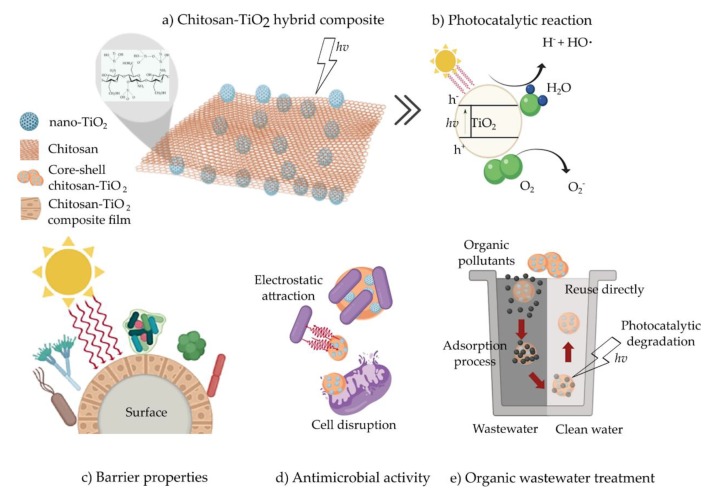
Chitosan–TiO_2_ hybrid composites and their applications (**a**) photocatalytic property, (**b**) potential barrier (**c**), antimicrobial (**d**), and wastewater (**e**) applications (figure created in Biorender.com).

**Table 1 materials-13-00811-t001:** Application of some polymers and proteins functionalized with TiO_2_ nanoparticles.

Material	Application	Ref.
Corn starch	Bioplastic with potential use as packaging material	[23]
Carboxymethyl cellulose containing miswak extract	Nanocomposite with potential use as food packaging	[24]
Sesame protein extract	Films for food active packaging applications and photo-decolorization purposes	[25]
Corn starch/PVA	Potential application in food and non-food industries as UV shielding packaging materials	[26]
Chitosan/potato-starch	Films with potential use as food packaging	[27]
Chitosan/starch	Films with potential use as food packaging	[28]
Cellulose	Removal of water pollutants	[29]
Bi_2_WO_6_-TiO_2_/corn starch	Films with ethylene scavenging activity, potential use for fruits and vegetables preservation	[30]
KC/X/G	Potential application in food and non-food industries as UV shielding packaging materials	[31]
Alginate	Removal of water pollutants	[32]
Potato starch	Films with potential use as food packaging	[33]
Amylose-halloysite composite	Composite with potential environmental applications as wastewater treatment	[34]
Wheat/cellulose	Films with antibacterial properties	[11]
Corn starch/PVA	Nanocomposite with potential use as packaging material	[35]
Alginate	Medical applications as tissue regenerator	[36]
Poly(L-lactic acid)	Potential use as drug delivery system	[37]
Whey protein	Biopolymer with potential use as packaging material	[38]
Hydrophilic polyurethane	Films with antibacterial properties	[39]

PVA: Polyvinyl alcohol; KC/X/G: mixture of K-Carrageenan (KC), xanthan gum (X), gellan gum (G).

**Table 2 materials-13-00811-t002:** Effect of TiO_2_ incorporation on chitosan (CS) matrix properties.

Parameter	Characterization Technique	Results	Ref.
Structural properties	SEM	Good dispersion of TiO_2_ nanoparticles into the CS film.	[40]
	EDX	TiO_2_ is uniformly distributed on the surface of the composite.	[41]
	AFM	Composite exhibited rough and porous surface.	[41]
	XRD	Characteristics peaks (2θ) for TiO_2_ (25.4°) and CS (20.4°) were reported on CS–TiO_2_ composite.	[5]
	UV-Vis	Composite exhibited a strong absorption range at 300–500 nm.	[42]
	Zeta–potential	CS–Ag–TiO_2_ coating exhibited good stability (z–potential of 33 mV) after 60 days of storage.	[43,44]
Textural properties	Ads–Des isotherm	Composite is classified as a macroporous material (isotherm Type II).	[45]
	SSA	Composite’s SSA is dependent of the CS:TiO_2_ ratio. A major presence of CS promotes a decrease in SSA.	[46]
	Pore volume	Decrease in pore volume in CS from 0.25 cm^3^ g^−1^ to 0.15 cm^3^ g^−1^ in CS–TiO_2_ composite.	[44]
	Pore size	TiO_2_ decreases the pore size of the composite.	[47]
Thermal properties	DSC	The presence of TiO_2_ enhances the thermal stability of CS.	[48,49,50,51,52,53]
	TGA	Composite film exhibited lesser degradation than a CS–film.	[4]
Optical properties	Color	CS film–forming solution turned whiter when TiO_2_ was added, affecting the color and transparency of the composite film.	[15]
	Light transmittance	Presence of TiO_2_ reduces the optical transmittance of the composite film.	[54,55,56]
Mechanical properties	Thickness	TiO_2_ promotes an increase in the composite film thickness.	[57]
	Young’s modulus	TiO_2_ improves the flexibility of the composite (increase of 11.8–fold in Young’s modulus).	[23]
	Tensile strength	TiO_2_ improves the elongation at break in 70%.	[54]
	Toughness	TiO_2_ enhances the toughness of composite (a six–fold increase).	[23]
	Viscosity	The viscosity of film-forming solution of CS–TiO_2_ is influenced by TiO_2_ content.	[58]
	Density	Composite–film exhibited low density (0.33 mg mm^−3^).	[59]
Vapor barrier properties	Water vapor barrier transmission rate	Incorporation of TiO_2_ on CS promotes a decrease in water vapor permeability.	[40]
	Oxygen barrier transmission rate	Presence of TiO_2_ significantly reduced the oxygen permeability. However, its effectiveness is TiO_2_ concentration-dependent.	[60]
	Water solubility	TiO_2_ did not influence in the water solubility behavior CS–TiO_2_ of composite.	[57]
Biodegradability	Swelling study	TiO_2_ controls the swelling capacity of CS film.	[45]
	Biodegradation rate	CS–TiO_2_ composite exhibited moderate biodegradability (<3%) and favorable time–dependent biodegradability (0.0521 mg mL^−1^ h^−1^ on day zero; 0.6 ng mL^−1^ h^−1^ post–covering).	[59]

SEM: Scanning electron microscopy; EDX: Energy Dispersive X-Ray Spectroscopy; AFM: Atomic force microscopy; XRD: X-ray diffraction; UV-Vis: UV-Vis diffuse reflectance spectroscopy; Ads-Des: Adsorption-desorption isotherms; SSA: Specific surface area; DSC: differential scanning calorimetry; TGA: thermogravimetric analysis.

**Table 3 materials-13-00811-t003:** FTIR band positions of chitosan–TiO_2_ composite.

Band Position	Assignment	Ref.
3450	OH bond of chitosan; it could exist an electrostatic interaction of N–H–O–Ti	[15,16]
3350	Combined peaks of the NH_2_ and OH group stretching vibrations	[69]
3300	Strong interaction between NH_2_ and OH with TiO_2_	[40,69]
2934	Asymmetrical stretching vibration of the C–H in CH_2_ and CH_3_ groups	[55,70]
2923–2872	C–H asymmetric and symmetric vibrations, TiO_2_–OH functional group	[8]
1735–1733	O–C–NH_2_ indicated the presence of titanates in the composite	[71]
1637–1715	N–H scissoring from the primary amine, it could exist an interaction of Ti^4+^ with –NH_2_	[50,68,72,73]
1577–1589	Angular deformation of N–H bonds	[51,74]
1538	Secondary amide (amide II), CH_2_ bending	[60]
1528–1534	C–N and C–N–H bending mode	[5,68]
1421	C–N axial deformation (amine group); C–O stretching (amide I)	[45,70]
1370–1420	C–O–C stretching bands, N = O vibrations, –NH deformation, CH_3_ group	[15,70,73,74]
1287	Ti–OH and Ti–O bonds	[15,70]
1029–1152	C–N bending vibrations and asymmetrical stretching vibrations of C–O–C glycosidic bonds, Ti–O–C bending mode, Ti–OH bond	[10,15,16,17,70,74,75]
600–900	Ti–O–Ti bond, asymmetric stretching mode of Ti–O, immobilization of TiO_2_ onto the CS matrix	[5,15,16,70,74]
385–600	Ti–O–C, it could exist an interaction of Ti Lewis site with -NH_2_ groups of chitosan chain	[42,50,69]

**Table 4 materials-13-00811-t004:** Antimicrobial activity of CS–TiO_2_ composite.

Microorganism	Material	Composition	Results	Ref.
*Staphylococus aureus, Escherichia coli, Salmonella Typhimurium, Pseudomonas aeruginosa, Aspergillus, Penicillium*	CS–TiO_2_ film	CS (2% w/v), nano–TiO_2_ (1% w/w)	The film exhibited antimicrobial activity against Gram–positive and Gram–negative bacteria, and fungi.	[40]
*Sthapylococcus aureus, Pseudomona aureginosa*	CS–TiO_2_:Ag composite	CS (1.5% w/v), TiO_2_ (0.1% w/v), Ag (1 mol L^−1^)	CS–TiO_2_:Ag exhibited major antibacterial activity than CS–TiO_2_ composite against *E. coli*, *S. aureus* and *P. aureginosa.*	[72]
*Sthapylococcus aureus*	CS–TiO_2_ membrane	CS (1% w/v), TiO_2_ (0.25% w/v)	Enhanced antibacterial activity against *S. aureus* by CS–TiO_2_ compared to CS films.	[4]
*Escherichia coli*	CS–Ag–TiO_2_ coating	CS (1% w/v), AgNO_3_ solution (1% w/v), 0.5% TiO_2_ NPs	CS–Ag–TiO_2_ exhibited higher antimicrobial activity (MIC of 0.38 µg mL^−1^) than the individual components (CS, Ag, TiO_2_) against *E. coli* (MIC > 4 µg mL^−1^).	[44]
*Escherichia coli*, *Sthapylococcus aureus, Candida albicans*	Ag–CS–TiO_2_ composite	CS–TiO_2_ (0.15 g) mixed in 40 mL of AgNO_3_ (Ag^+^ 200 mg L^−1^)	Composite exhibited favorable antimicrobial activity against *E. coli*, *S. aureus* and *C. albicans*, without significant losses of its activity even after five consecutive cycles.	[6]
*Escherichia coli, Sthapylococcus aureus, Salmonella enterica* ser. Typhimurium	CS–TiO_2_ film	CS (2% w/v), TiO_2_ (0.1% w/v) NPs	Films were effective in reducing the microbial concentration in liquid culture for *S. aureus, E. coli* and *Salmonella enterica* ser. Typhimurium, but effectiveness was dependent on the strain and TiO_2_ content.	[15]
*Escherichia coli*, *Sthapylococcus aureus, Candida albicans, Aspergillus niger*	CS–TiO_2_ film	CS (2.5% w/v), TiO_2_ NPs (2.5% w/v)	CS–TiO_2_ films exhibited photocatalytic antimicrobial activity against *E. coli*, *S. aureus*, *C. albicans*, and *A. niger.*	[54]
*Salmonella choleraesuis*	TiO_2_ on CS beads and activated carbon	CS (2% w/v), nano–TiO_2_ (5% w/v)	TiO_2_ coated on activated carbon and chitosan beads served as a strong anti-bacterial agent against *S. choleraesuis* subsp.	[78]
*Escherichia coli, Staphylococcus aureus*	CS–TiO_2_:Cu composite	CS (1% w/v), nano–TiO_2_:Cu (0.2 mg mL^−1^)	CS–TiO_2_: Cu exhibited antimicrobial activity against *E. coli* and *S. aureus*, it was enhanced in 200% in presence of UV-light.	[10]
*Escherichia coli*	TiO_2_/CS/CMM coating layer	CS (1,3,5% w/t), TiO_2_ (2%), CMM	TiO_2_/CS exhibited significant reduction (93%) in viable cells *E. coli* in viable cells after 24 h.	[64]
*Escherichia coli, Aspergillus niger, Candida albicans*	CS/TiO_2_ emulsion deposited on gauzes	CS (0.1% w/v), TiO_2_ (0.05 g), gauze 25 cm^2^	Gauze treated with chitosan/nano–TiO_2_ composite emulsion showed antibacterial activities against *E. coli*, *A. niger* and *C. albicans.*	[79]
*Escherichia coli, Candida albicans, Aspergillus niger*	CS–Fe–TiO_2_ coatings	Chitosan (0.1% w/v), Fe–TiO_2_ (0.05 g)	Antimicrobial activity against *E. coli*, *C. albicans* and *A. niger* under visible light irradiation.	[41]
*Aspergillus niger, Bacillus subtilis*	CS–GO–TiO_2_ coating	CS (1% w/v), GO:TiO_2_ ratio 1:2	Self-assembled film of GO–CS with nano–TiO_2_ exhibited high antibacterial activity against biofilm forming *A. niger* and *B. subtilis.*	[80]
*Escherichia coli, Staphylococcus aureus, Aspergillus niger*	CS–TiO_2_ incorporated in cotton fibers	CS (1% w/v), TiO_2_ (0.5 g), cotton fiber 250 cm^2^	More than 99% of *E. coli*, *S. aureus* and *A. niger* of viable cells were inactivated.	[42]
*Staphylococcus aureus, Escherichia coli, Klebsiella pneumoniae*	CS–TiO_2_ composite	CS:TiO_2_ (2:1)	Moderate antibacterial activity against *S. aureus*, but no for *E. coli* and *K. pneumoniae.*	[68]
*Escherichia coli*	CS–TiO_2_ composite	CS (1% w/v), TiO_2_ (1% w/v)	CS–TiO_2_ exhibited a complete inactivation of *E. coli* (100% of reduction) in comparison with a CS alone (7.5%) after 24 h of exposure.	[74]
*Escherichia coli, Salmonella enteritidis, Staphylococcus aureus*	Zein/CS/TiO_2_ film	CS (0.12 g), Zein (2 g), TiO_2_ (0.25% w/w)	Antimicrobial activity of Zein/CS was improved by the incorporation of TiO_2_.	[55]
*Xanthomonas oryzae pv. Oryzae*	CS–TiO_2_ composite	CS (1% w/v), TiO2 (0.50% w/w)	CS–TiO_2_ exhibited enhanced antibacterial activity compared with the individual components.	[9]
*Escherichia coli, Staphylococcus aureus*	TiO_2_–CS–PVA composite	TiO_2_ (0.1 g), CS (0.5 g), PVA solution (10%), mass ratio 1:5:20	Composite exhibited enhanced antibacterial activity against *E. coli* and *S. aureus* compared with the individual components.	[81]

CS: chitosan; CMM: Cellulose microfibers mat; GO: graphene oxide.

**Table 5 materials-13-00811-t005:** Photocatalytic activity of a CS–TiO_2_ composite for different water pollutant degradation.

Pollutant	Material	* Composition	Relevant Results	Ref.
Methyl orange	CS–TiO_2_ composite	Different weight ratios of TiO_2_ and CS were evaluated (75:25, 50:50, 25:75) being the composite 1% of the total volume of polluted water.	TiO_2_–CS composite (75:25 w/w) showed great degradation efficiency against methyl orange dye and the recycling outcome material shows effective stability.	[16]
Thymol violet	TiO_2_/CS/CMM coating layer	CS (1,3,5% w/t), TiO_2_ (2%), CMM	TiO_2_/CS/CMM composites could adsorb thymol violet for its removal from water.	[64]
Methyl orange/Congo red	CS/PVA/Na–Titanate/TiO_2_ composites	CS–PVA ratio 60:40, 80:20, 90:10, TiO_2_ (1% w/v)	100% of methyl orange removal was obtained under UV irradiation, and 99% for Congo red removal.	[48]
Cu(II) and Pb(II) heavy metal ions	CS–TiO_2_ composite	CS (7% w/v), TiO_2_ (1% w/v)	CS–TiO_2_ exhibited higher potential for metal ions sorption compared with CS.	[17]
Methylene blue	CS–TiO_2_ nanohybrid	CS (1% w/v), TiO_2_ (1% w/v)	Exhibited high photocatalytic activity degradation of methylene blue dye under UV–light illumination even after five cycles of reuse.	[74]
Cr(VI)	CS–TiO_2_ beads	CS (1% w/v), TiO_2_ (1% w/v)	CS–TiO_2_ composite exhibited high reduction of Cr(VI) in water in comparison with CS.	[75]
Methyl orange	CdS/TiO_2_/CS coating	CS (2% w/v), TiO_2_ (0.3% w/w), CdCl_2_ (0.912% w/v)	CdS/TiO_2_/CSC exhibited enhanced photocatalytic activity under simulated solar light irradiation and represents a suitable and promising photocatalyst for effective decolorization treatment of dye–containing effluents.	[84]
Methyl orange	TiO_2_/ZnO/CS composite	CS (2% w/v), TiO_2_ (0.2 g), ZnO (1.17 g of zinc acetate)	Exhibited high photocatalytic activity for methyl orange degradation (97%) under simulated solar radiation.	[77]
Methyl orange	CS–TiO_2_ composite	CS (1 g in 36 mL of acetic acid), TiO_2_ (0.025 g)	Enhanced photocatalytic selectivity for methyl orange compared with CS, and could be reused up to 10 cycles without desorption and regeneration while preserving 60% of its photocatalytic efficiency.	[6]
Cr(VI)	CS–TiO_2_ composite	CS (3.22% w/v), TiO_2_ solution	The CS–TiO_2_ composite was quite effective for adsorption and detoxification of Cr(VI) in water with a maximum adsorption capacity of 171 mg g^−1^ for Cr(VI).	[50]
As	CS–TiO_2_ composite	CS (1 g), TiO_2_ (0.42g g^−1^ CS)	Exhibited good Arsenic removal from water, but the effectiveness is TiO_2_ concentration dependent.	[85]
Congo red	CS–TiO_2_ glass photocatalyst	CS (0.83% w/v), TiO_2_ (0.83% w/v)	The study suggests a new method that has the advantages of photodegradation and adsorption in the abatement of various wastewater pollutants.	[8]
Methyl orange	CS–TiO_2_ composite	CS (0.5 g 25 mL^−1^), TiO_2_ (0.2 g 25 mL^−1^)	Exhibited degradation of methyl orange and adsorption on Ni^2+^ ions.	[86]
Cd, 2,4-DCP	MICT composite	CS (2.5 g 500 mL^−1^), TiO_2_ (1 g TiO_2_), Fe_3_O_4_ (1.25 g)	The composite is effective for cadmium adsorption (256 mg g^−1^) and 2,4–dichlorophenol degradation (98%), and could be used up to five cycles with preserving 69% of its adsorption properties.	[73]
Methyl orange	TiO_2_–CS–rGO	CS (2% w/v), 4 mL of TiO_2_ suspension (50 mg mL^−1^), rGo (1% w/w)	Hybrid composite showed photocatalytic degradation (97%) of methyl orange.	[58]
Methylene blue	CS–TiO_2_ composite	CS (1% w/v), TiO_2_ (1 g in 20 mL of CS solution)	Composite is effective for methylene blue (100%) degradation under UV–light, but the effectiveness was catalyst concentration–dependent.	[45]
As(III), As(V)	CS–TiO_2_:Cu composite	CS (1% w/v), TiO_2_ (0.6% w/w), Cu (0.7% w/w)	The composite showed good photo–oxidation and selective removal for As(III) and As(V).	[87]
Methyl orange	CS–TiO_2_ composite	CS (1% w/v), TiO_2_ (2% w/w)	CS is an excellent support for TiO_2_ immobilization, which exhibited a complete photodegradation of methyl orange and alizarin red S after 3 h of treatment.	[88]
4-NPh	CS–TiO_2_:Ag composite	CS (1% w/v), TiO_2_ (0.4% w/v), Ag^+^ (200 mg L^−1^)	Incorporation of Ag+ ions into the surface of CS–TiO2 composite improved the catalytic activity in the reduction of 4–NPh to 4–APh (100% in 120 min) and preserve the catalytic activity in five continuous cycles.	[89]
Ni, Cd, Cu, Hg, Mn and Cr heavy metal ions	CS–HC–TiO_2_ composite	CS (1.5 g in 56 mL), TiO_2_ (0.04 g), HC (6 g)	Composite is effective for heavy metal ions (Ni, Cd, Cu, Hg, Mn and Cr) adsorption from aqueous solution.	[90]
As(III), As(V)	TiO_2_/feldspar(FP)-embedded in CS beads	CS (1% w/v), TiO_2_ (0.5 g/1g CS), FP (0.5 g^−1^ g^−1^ CS)	The composite exhibited good adsorption properties for arsenite and aresenate removal from aqueous solution.	[83]
Cd(II)	Cd sensor	CS–TiO_2_ composite film onto a glassy carbone electrode	The modified electrode exhibited a detection limit of 2 × 10^−10^ mol L^−1^ Cd for 180 s accumulation.	[91]
Acid fuchsin	TiO_2_/CS/PNIPAAm) composite hydrogel	CS (0.15 g dissolved in 10 mL of 1% acetic acid solution), TiO_2_ (0.5 g), PNIPAAm (0.5 g)	Composite hydrogel exhibited high efficiency photocatalytic degradation for acid fuchsin, and the removal reached 90% after 160 min.	[92]
Cd(II)	EDTA/CS/TiO_2_ composite	CS (0.2 g), TiO_2_ (0.1 g)	Composite exhibited high adsorption properties from Cd (II) removal and high phenol degradation efficiency.	[93]
Methyl orange	CS–TiO_2_ composite	CS (2% w/v), TiO_2_ (NI)	A complete degradation of methyl orange under UV exposition during 90 min was obtained.	[94]
2-4-DCP	CS/ACF/TiO_2_ membrane	CS (1% w/v), TiO_2_ (NI), ACF (NI)	The membrane exhibited high efficiency on 2–4–DCP removal from aqueous solution.	[95]
Methyl orange	CS/PVA/TiO_2_ composite	CS (7 wt. %.), PVA (8 wt. %.), TiO_2_ (1 wt. %.)	Deacetylation degree of chitosan have an impact on methyl orange removal.	[48]

* CS: chitosan was dissolved in acetic acid solution; PVA: Poly(vinyl alcohol); CMM: Cellulose microfibers mat; MICT: Magnetic ion-imprinted chitosan-TiO_2_; rGO: reduced graphene oxide; HC: Hemicellulose; ACF: activated carbon fiber; MO: methyl orange; 2–4–DCP: 2,4–dichlorophenol; NI: no information.

**Table 6 materials-13-00811-t006:** Potential biomedical application of a CS–TiO_2_ composite.

Application	Material	Composition	Relevant Results	Ref.
Wound healing	CS–TiO_2_ membrane	CS (1% w/v), TiO_2_ (0.25% w/w)	Membranes allow proliferation, survival, and decreased oxidative stress and apoptosis of L929 cells.	[4]
Immuno-biosensors	Au/CS/TiO_2_–graphene composite	CS (1 mg mL^−1^), TiO_2_-Gr (1 mg), Au (NI)	Biosensor exhibited good bioactivity, sensitivity (0.1–300 ng mL^−1^) and selectivity for α-fetoprotein detection. Possible applications on the detection of other antigens or biocompounds.	[98]
Wound healing	CS–TiO_2_ with collagen artificial skin (NTCAS)	CS (2%), TiO_2_ (0.40%)	In an animal model, NTCAS had better outcomes with regard to integrated wound healing than a commercial product.	[59]
Glucose biosensor	Ppy–CS–TiO_2_ nanocomposite film	CS (50 mg mL^−1^), TiO_2_ (NI), Ppy (NI)	Biosensor showed good sensitivity over linear range of 1–14 mM with detection limit of 614 µM for glucose (R^2^ = 0.989).	[99]
Glucose biosensor	TiO_2_–CN–CS composite functionalized with nano–Au	2 mg TiO_2_-CN was dispersed in 2 mL CS solution (1 mg mL^−1^), Au (NI)	Biosensor showed good response performance to glucose with a linear range of 6 µM to 1.2 mM with a detection limit of 0.1 µM glucose.	[100]
Tissue engineering applications	chitin–CS/nano–TiO_2_ composite scaffolds	CS (2% w/v), TiO_2_ (2% w/w)	No cytotoxic effects on MG-63, L929, and hMSCs cell lines were observed.	[47]
Wound healing	CS–Pectin–TiO_2_ composite	CS:Pectin (1:1), TiO_2_ (0.001% w/w)	The wounds treated with CS-Pectin-TiO_2_ dressing material healed faster than CS-treated and gauze.	[49]
Wound healing	TiO_2_–CS–ECM	CS (8.6% w/v), TiO_2_ (1% w/w), ECM sheets	TiO_2_-CS-ECM exhibited wound healing acceleration effects.	[101]
Tissue engineering applications	CS–TiO_2_ composite	CS (1% w/v), TiO_2_ (2:1)	No cytotoxic effect of the composite on a gastric carcinoma human cell line. The preparation method has a remarkable effect on composite biocompatibility.	[46]
Regulation of osteoblast bioactivity	CS–gelatin composite coating on ICA–modified TiO_2_ nanotubes	CS (10 mg mL^−1^), gelatin (NI), TiO_2_ (0.5 mg mL^−1^), ICA (NI)	Composite promote osteoblast proliferation and up-regulation on the expression of bone-related genes (osteopontin, type I collagen, and osteoprotegerin) while down-regulating RANKL mRNA expression.	[102]
Drug delivery system	CS/DOP/TiO_2_ composite	CS (1% w/v), DOP (0.01–0.04% w/v), TiO_2_ (30% mass ratio)	Incorporation of TiO_2_ on CS/DOP composite considerably reduces the drug release (16 h) in comparison with CS/DOP system (10 min).	[103]
Bone regeneration	TiO_2_–CS–H4S composite	TiO_2_-CS-CH4S molar ration 2:1:0.125	The composite exhibited high bioactivity and biocompatibility with human MG–63 cell line.	[67]
Medical dressing	TiO_2_–CS–PVA	TiO_2_ (0.1 g), CS (0.5 g), PVA solution (10%), mass ratio 1:5:20	Composite did not show toxicity against L929 cell line.	[9]

NI: No information; Ppy: polypyrrole; ICA: icariin; DOP: Dopamine; CH4S: chondrotoin 4-sulphate; CN: Carbon nanotubes; ECM: electrospun chitosan membrane; PVA: Polyvinyl alcohol.

**Table 7 materials-13-00811-t007:** Potential food applications of CS–TiO_2_ composite.

Food	Material	Composition	Relevant Results	Ref.
Cherry tomatoes	CS–TiO_2_ film	CS (1% w/v), TiO_2_ (1% w/v) of TiO_2_	Cherry tomatoes packaged with CS–TiO_2_ nanocomposite film had lower quality changes (firmness, weight loss, color, TSS, lycopene and AA content, and concentration of ethylene and CO_2_) and delayed the ripening process.	[5]
Cantaloupe rind	CS–Ag–TiO_2_ coating	CS (1% w/v), AgNO_3_ solution (1% w/v), TiO_2_ (0.5% w/v)	The CS–Ag–TiO_2_ exhibited good adherence on cantaloupe rind surface.	[44]
Red grape	CS–TiO_2_ film	CS (2.5% w/v), TiO_2_ (2.5% w/v)	The CS–TiO_2_ film prevented microbial infection and increased the shelf life of red grape to 36 days at 37 °C.	[54]
Strawberries and Mangoes	CS–GO–TiO_2_ coating	CS (1% w/v), GO:TiO_2_ ratio 1:2	Coated fruits exhibited less than 5% weight loss and maintained color attributes compared to uncoated fruits. PPO activity was diminished in coated fruits.	[80]
Stauntonive fruit	CS–TiO_2_ coating	CS (1% w/v), TiO_2_ (0.03% w/w)	The fruit treated with the composite coating exhibited a reduction on the CO_2_ transmission coefficient compared to the CS–treated, without significant changes in quality parameters during 45 days of storage.	[105]
*Gingko biloba* seeds	CS–TiO_2_ coating films	CS (1% w/v), TiO_2_ (0.02% w/v)	The composite preserved the quality parameters (firmness and antioxidant activity) of gingko seeds and prevented the mildew apparition.	[106]

TSS: Total soluble solids; AA: Ascorbic acid; PPO: Polyphenol oxidase; SOD: superoxide dismutase.

**Table 8 materials-13-00811-t008:** Textile and anti-corrosive applications of CS–TiO_2_ composites.

Application	Material	Composition	Relevant Results	Ref.
Textile	CS–TiO_2_ coated onto cotton	CS (10 g in 50 wt. % NaOH solution), 1 mL of TiO_2_ (3.28 mmol)	The treated cotton (CS–TiO_2_) showed excellent protection against UV radiation in comparison to a cotton–TiO_2_ system.	[56]
Textile	TiO_2_–CS and CA	TiO_2_:CS:CA relation 1:3:2 wt. %	The treated wool fabrics cured with CA and TiO_2_–CS showed good protection against UV–radiation.	[108]
Metal corrosion resistance	CS–TiO_2_ composite	CS:TiO_2_ molar ratio 1:1	CS–TiO_2_ composite improved the corrosion resistance of aluminum metal compared with chitosan.	[109]
Metal corrosion resistance	CS–TiO_2_ coating	CS (0.5 g L^−1^), TiO_2_ (3 g L^−1^)	Hybrid coating improved the corrosion resistance of X2CrNiMo17–12–2 stainless steel.	[110]

CS: Chitosan; GO: Graphene oxide; CA: citric acid.

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
