# Peer review of "Chitosan-TiO2: A Versatile Hybrid Composite"

_materials, 2020, doi:10.3390/ma13040811_

Round 1

Reviewer 1 Report

Review of the manuscript: “Chitosan-TiO2: A versatile organometallic composite”

This manuscript reviews most of the research works in chitosan/TiO2composites, touching different aspects (synthesis, applications in the field of antimicrobial materials, environmental remediation, textile and anti-corrosion coatings). Overall, the work of summarizing and discussing the most relevant applications of these materials has been properly done, thus the manuscript could be accepted for publication.

The following points should be addressed:

The term “organometallic”, which is utilized in the manuscript starting from the title, is not correct and somewhat misleading. The definition of organometallic entails the presence of at least one chemical bond between the organic and inorganic moieties. The examples discussed in the review refer to composites. In most of the cases the only types of interaction between TiO2 and chitosan are non-covalent. I would suggest the replacement of the term “organometallic” with “hybrid”. In the general description of hybrid nanocomposites (paragraph 2), further references to the application of alginate/TiO2composites, which are closely related to the chitosan/TiO2systems, could be added to guide the readers through a comparative analysis (see, for example, Journal of environmental chemical engineering (2017) 5 (2), 1763-1770). Table 3 reports the assignments to the bands of FTIR spectra of chitosan/TiO2 The visualization of a typical FTIR spectrum could be helpful to associate those assignments to the experimental data and provide the researchers with a direct reference to compare their own spectra. Row 316: there is a typo: grapheme should be graphene Table 5: first row (ref 16): the TiO2% is missing.

Author Response

RESPONSE TO REVIEWER´S COMMENTS

Dear editor and reviewers:

Thank you very much for your accurate comments. We have done our best to follow up on all your recommendations. Note: Changes in the revised manuscript are in red letters.

Reviewer 1:

This manuscript reviews most of the research works in chitosan/TiO2 composites, touching different aspects (synthesis, applications in the field of antimicrobial materials, environmental remediation, textile and anti-corrosion coatings). Overall, the work of summarizing and discussing the most relevant applications of these materials has been properly done, thus the manuscript could be accepted for publication.

The following points should be addressed:

The term “organometallic”, which is utilized in the manuscript starting from the title, is not correct and somewhat misleading. The definition of organometallic entails the presence of at least one chemical bond between the organic and inorganic moieties. The examples discussed in the review refer to composites. In most of the cases the only types of interaction between TiO2 and chitosan are non-covalent. I would suggest the replacement of the term “organometallic” with “hybrid”.

“Organometallic” term was changed by “hybrid” term in all document.

In the general description of hybrid nanocomposites (paragraph 2), further references to the application of alginate/TiO2 composites, which are closely related to the chitosan/TiO2 systems, could be added to guide the readers through a comparative analysis (see, for example, Journal of environmental chemical engineering (2017) 5 (2), 1763-1770).

We consider that an extensive comparison of other biopolymer-based hybrid composites in combination with TiO2, is discussed in section 2, lines 87–177. Moreover, the suggested reference was included in Table 1, and some examples of hybrid composites were added in lines 41–43.

Table 3 reports the assignments to the bands of FTIR spectra of chitosan/TiO The visualization of a typical FTIR spectrum could be helpful to associate those assignments to the experimental data and provide the researchers with a direct reference to compare their own spectra.

A typical FTIR spectrum of TiO2, chitosan, and chitosan-TiO2 hybrid composite was added in the document (Figure 2a). CS-TiO2 hybrid composite film was prepared according to the methodology proposed by Siripatrawan and Kaewklin [40], using 1g of chitosan with a medium molecular weight dissolved in 1% acetic acid solution, nano-TiO2 at 1% of chitosan relative weight, which were previously characterized by Anaya-Esparza et al. [14], and 15 % of glycerol of the total of solids.

Anaya-Esparza, L.M.; Montalvo-González, E.; González-Silva, N.; Méndez-Robles, M.D.; Romero-Toledo, R.; Yahia, E.M.; Pérez-Larios, A. Synthesis and characterization of TiO2-ZnO-MgO mixed oxide and their antibacterial activity. Materials 2019, 12, 698.

Siripatrawan, U.; Kaewklin, P. Fabrication and characterization of chitosan-titanium dioxide nanocomposite film as ethylene scavenging and antimicrobial active food packaging. Food Hydrocoll. 2018, 84, 125–134.

Row 316: there is a typo grapheme should be graphene.

Typo “Grapheme” was corrected by “graphene” and all document was revised and corrected.

Table 5: first row (ref 16): the TiO2% is missing.

Concentration of TiO2 was added in Table 5, Saravanan et al. [16] evaluated the effect of different weight proportions of TiO2 and CS (75:25, 50:50, 25:75) nanocomposite on methyl orange dye degradation.

Reviewer 2 Report

Thank you authors for this interesting rewiev. The review is very complete and  I found it very usefull, especially the chapter "Applications". I can recomend it but after some minor questions that I would like to discuss.

Tables are very usefull, but I miss more figures, adaptations from the most relevant papers, especially in the "applications chapter". I suggest you choose at least 3 of the most relevant paper for you in the area and adapt some of the their best figures and explain them. I found the conclusions very poor. They are only a short summary of the benefict of this kind of material. But I would like to read the current opinion of the authors (there is only a short phrase "However, additional ......") In your opinion, what is the future of this kind of materials, what is the most available applications, what possibilities have more free room for research development.... In the Abstract, sentence of line 33-35 is very similar to sentence in line 29 (UV-barrier propierties...), only change the application. In my opinion you could reformulate that paragraph to put all together. Line 64 and in line 67. I can not consider TiO2 as mesoporous material with high surface area. There are many synthesis protocols to get mesoporous titania, but primarly titania is not porous. If you are only talking about mesoporous TiO2, please, reformulate this paragrahp to be more easy to understand.

Author Response

RESPONSE TO REVIEWER´S COMMENTS

Dear editor and reviewers:

Thank you very much for your accurate comments. We have done our best to follow up on all your recommendations. Note: Changes in the revised manuscript are in red letters.

Reviewer 2:

Thank you, authors, for this interesting review. The review is very complete and I found it very useful, especially the chapter "Applications". I can recommend it but after some minor questions that I would like to discuss.

Tables are very useful, but I miss more figures, adaptations from the most relevant papers, especially in the "applications chapter". I suggest you choose at least 3 of the most relevant paper for you in the area and adapt some of their best figures and explain them.

A Schematic diagram of Chitosan-TiO2 hybrid composite and the main applications was added as Figure 3.

I found the conclusions very poor. They are only a short summary of the benefit of this kind of material. But I would like to read the current opinion of the authors (there is only a short phrase "However, additional ......") In your opinion, what is the future of this kind of materials, what is the most available applications, what possibilities have a more free room for research development.

The conclusion of the review was improved

In the Abstract, the sentence of line 33-35 is very similar to a sentence in line 29 (UV-barrier properties...), only change the application. In my opinion you could reformulate that paragraph to put all together.

The sentence was modified as suggested

Line 64 and in line 67. I can not consider TiO2 as mesoporous material with high surface area. There are many synthesis protocols to get mesoporous titania, but primarily titania is not porous. If you are only talking about mesoporous TiO2, please, reformulate this paragraph to be more easy to understand.

The reviewer is right, the mesoporous character of TiO2 and its specific surface area are mainly dependent on the synthesis method. In this context, we prefer to skip the mesoporous term in the lines 64 – 67, due to not all authors included or specify the porosity classification of the materials used in their researchers, as the IUPAC established.

International Union of Pure and Applied Chemistry. Reporting physisorption data for gas/solid system with special reference to the determination of surface area and porosity. IUPAC, 1985, 57, 603-619.
